# IMAGINE THAT! LEVERAGING EMERGENT AFFORDANCES FOR TOOL SYNTHESIS IN REACHING TASKS

## ABSTRACT

In this paper we investigate an artificial agent's ability to perform task-focused tool synthesis via imagination. Our motivation is to explore the richness of information captured by the latent space of an object-centric generative model - and how to exploit it. In particular, our approach employs activation maximisation of a task-based performance predictor to optimise the latent variable of a structured latent-space model in order to generate tool geometries appropriate for the task at hand. We evaluate our model using a novel dataset of synthetic reaching tasks inspired by the cognitive sciences and behavioural ecology. In doing so we examine the model's ability to imagine tools for increasingly complex scenario types, beyond those seen during training. Our experiments demonstrate that the synthesis process modifies *emergent*, task-relevant object affordances in a targeted and deliberate way: the agents often specifically modify aspects of the tools which relate to meaningful (yet implicitly learned) concepts such as a tool's length, width and configuration. Our results therefore suggest, that task relevant object affordances are implicitly encoded as *directions* in a structured latent space shaped by experience.

## 1 INTRODUCTION

Deep generative models are gaining in popularity for unsupervised representation learning. In particular, recent models like MONet (Burgess et al., 2019) have been proposed to decompose scenes into object-centric latent representations (cf. Greff et al., 2019; Engelcke et al., 2019). The notion of such an object-centric latent representation, trained from examples in an unsupervised way, holds a tantalising prospect: as generative models naturally capture factors of variation, could they also be used to expose these factors such that they can be modified in a task-driven way? We posit that a task-driven traversal of a structured latent space leads to *affordances* emerging naturally as directions in this space. This is in stark contrast to more common approaches to affordance learning where it is commonly achieved via direct supervision or implicitly via imitation (e.g. Tikhanoff et al., 2013; Myers et al., 2015; Liu et al., 2018; Grabner et al., 2011; Do et al., 2018).

The setting we choose for our investigation is that of tool synthesis for reaching tasks as commonly investigated in the cognitive sciences. The ability to make and use tools has been a cornerstone of humanity's cognitive, technological and cultural development. It has long been hailed as a defining characteristic of intelligence (Ambrose, 2001). Consequently, the ability of artificial agents to select and use appropriate tools has received significant attention in, for example, the robotics community (e.g. Sinapov & Stoytchev, 2007; Tikhanoff et al., 2013; Toussaint et al., 2018; Xie et al., 2019). A very limited number of works such as Wicaksono (2017) also consider tool making and typically deploy strong inductive biases such as simulations in which the agent adjusts numerical parameters given a set of design constraints. However, while much stands to be gained by endowing robots with an aptitude for tool use, more recent evidence in the cognitive sciences suggests that habitual tool use *cannot* in and off itself be taken as a sign of intelligence (e.g. Ambrose, 2001; Emery & Clayton, 2009). Some animals, which are recognised as prodigious users of tools,

do not show any sign of understanding either cause or effect or, indeed, the difference between appropriate and inappropriate tools (e.g. Ambrose, 2001). Instead, relatively recent results suggest that tool selection and manufacture – especially once demonstrated – is a significantly easier task than tool *innovation*: the step, prior to manufacture, of *imagining* the type of tool suitable for a particular task (Beck et al., 2011). We posit that it is tool innovation which ultimately unlocks the full potential of tool manufacture and, thus, creative tool use.

In order to demonstrate that a task-aware, object-centric latent space encodes useful affordance information we require a mechanism to train such a model as well as to purposefully explore the space. To this end we propose an architecture in which a task-based performance predictor (a classifier) operates on the latent space of an object-centric generative model (see Figure 1). During training the classifier is used to provide an auxiliary objective, aiding in shaping the latent space. Importantly, however, during test time the performance predictor is used to guide exploration of the latent space via activation maximisation (Erhan et al., 2009; Zeiler & Fergus, 2014; Simonyan et al., 2014), thus explicitly exploiting the structure of the space. While our desire to affect factors of influence is similar in spirit to the notion of disentanglement, it contrasts significantly with approaches such as $\beta$-VAE (Higgins et al., 2017), where the factors if influence are effectively encouraged to be axis aligned. Our approach instead relies on a high-level auxiliary loss to discover the direction in latent space to explore.

Our experimental design is inspired by a long history in the cognitive sciences and behavioural ecology of investigating reaching tasks which require tool use (e.g. Kohler, 1925; van Leeuwen et al., 1994; Chappell & Kacelnik, 2002; 2004; Emery & Clayton, 2009; Beck et al., 2011). The results demonstrate that artificial agents are indeed able not only to select – but to *imagine* – an appropriate tool for a variety of reaching tasks. The dataset consisting of 52,000 synthetic example reaching tasks will be made available to the community.

To the best of our knowledge, ours is the first work to demonstrate an artificial agent's ability to imagine, or synthesise, images of tools appropriate for a given task via optimisation in a structured latent embedding. Similarly, while activation maximisation has been used to visualise modified input images before (e.g. Mordvintsev et al., 2015), we believe ours to be the first to effect deliberate manipulation of factors of influence by chaining the outcome of a task predictor to the latent space, and then decoding the latent representation back into an image. Beyond the application of tool synthesis, in demonstrating that object affordances can be viewed as *directions* in a structured latent space as well as by providing a novel architecture adept at deliberately manipulating interpretable factors of influence we believe our work to provide novel perspectives on affordance learning and disentanglement.

## 2 RELATED WORK

The idea of an *affordance*, which describes a potential action to be performed on an object (e.g. a doorknob *affords* being turned), goes back to Gibson (1977). Together, robotics and computer vision boast a rich literature on learning task-affordances (e.g. Tikhanoff et al., 2013; Mar et al., 2015; Stoytchev, 2005), which includes work on learning multiple attention-like affordance masks (provided labelled data) for common objects such as kitchen utensils (Do et al., 2018; Myers et al., 2015). To learn affordances without supervision, another research thread has employed simulations in virtual environments. Grabner et al. (2011) is an example in which a simulated humanoid was re-positioned, via trial-and-error, to find all of the surfaces in a 3D scene in which it is possible to sit. Other studies have successfully learned affordances through imitation (e.g. Liu et al., 2018).

Closely related to the the affordance literature is a mature body of research dealing with almost all aspects of robot tool use. Tasks addressed include reaching (Jamone et al., 2015), grasping (Takahashi et al., 2017), pushing (Stoytchev, 2005) and hammering (Fang et al., 2018). This research commonly depends on a more traditional pipeline approach, consisting of tool-recognition (Tikhanoff et al., 2013; Fang et al., 2018), tool-selection (Xie et al., 2019; Saito et al., 2018), planning (Toussaint et al., 2018) and execution (Tikhanoff et al., 2013; Fang et al., 2018; Stoytchev, 2005; Jamone et al., 2015).

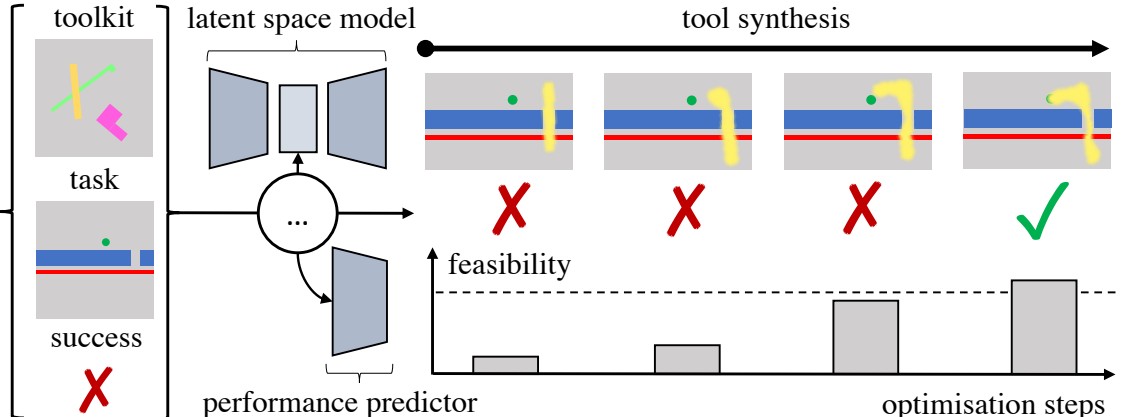

Figure 1: Innovating tools in a simulated reaching task. The model is trained on data triplets comprising the toolkit, the task environment, and an indicator of success (left). The toolkit is an image of stick-like tools (here in yellow, light green, and pink). The goal of the task is to determine if any tool can reach the dark green target whilst touching the red boundary and avoiding any blue obstacles. When no object in the toolkit satisfies these geometric constraints, our model (TasMON) is able to take a tool from the toolkit and *imagine* suitable modifications until it does satisfy these constraints (right). Here we see TasMON imagine the yellow stick transforming into a hook with a thin enough handle to both pass through the gap between the blue obstacles and reach the dark green target, all whilst still crossing over the red boundary.

Recently, the advent of deep generative models has led to an active research area using *world models* (Ha & Schmidhuber, 2018; Lesort et al., 2018; Nair et al., 2018), in which an artificial agent can train itself on tasks using a kind of imagination. Our work is directly inspired by this, though we approach imagination modelling in a different way. In particular, we propose an architecture in which a generative model for unsupervised scene decomposition (Burgess et al., 2019) is paired with the use of activation maximisation (e.g. Erhan et al., 2009; Zeiler & Fergus, 2014; Simonyan et al., 2014) of an auxiliary, task-driven loss-signal back into the generative model's latent representations to drive the imagination process.

## 3 DATASET

To investigate tool imagination, we designed a set of simulated reaching tasks with clear and controllable factors of influence. Each task image is comprised of a green target button, a red line delineating the workspace area, and, optionally, a set of blue obstacles. We also vary the goal location and the sizes and positions of the obstacles. For each task image, we provide a second image depicting a set of 'tools' – straight sticks, L-shaped hooks, and J-shaped claws – with varying dimensions, shapes, colours, and poses. Given a pair of images (i.e. the task image and the toolkit image), the goal is to select a tool for a given task scene that can reach the target (the green dot) whilst avoiding obstacles (blue areas) and remaining on the exterior of the workspace (i.e. behind the red line). Depending on the task image, the applicability of a tool is determined by different subsets of its attributes. For example, if the target button is unobstructed, then any tool of sufficient length will satisfy the constraints (regardless of its width or shape). However, when the target is hidden behind a corner, or only accessible through a narrow gap, an appropriate tool also needs to feature a long-enough hook, or a thin-enough handle, respectively. By design, the colour of a tool does not influence its applicability and is introduced as a nuisance factor. As depicted in Figure 2, we have designed eight scenario types to study these factors of influence in isolation and in combination. Importantly, the scenario types are designed to enable an investigation into our approach's performance on *in-sample* tasks, *interpola-*

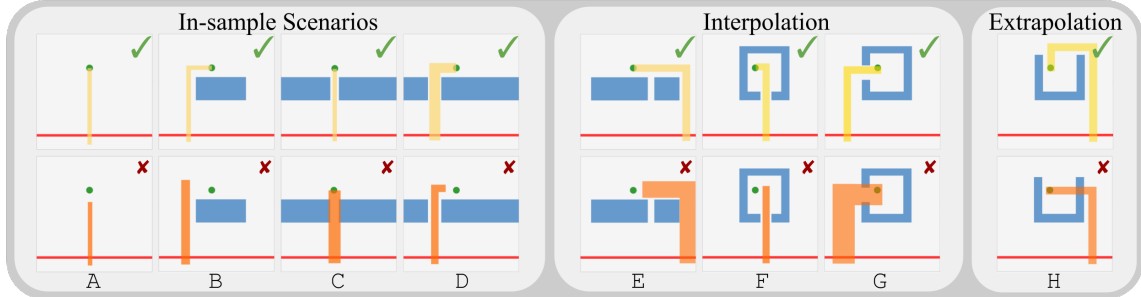

Figure 2: Illustrative examples of tasks in our dataset. Each task has a green target, a red line indicating the movement limits of the robot, and potentially some blue obstacles. We train on the first four *in-sample scenario types*: A − D. During evaluation, we also include *interpolation scenario types*: E − G, which can be solved by the tools seen in training data, as well as an *extrapolation scenario type*: H, which can only be solved by claw tools never seen in the training data. We choose these scenarios to set constraints on the dimensions of an appropriate tool; for example, in scenario C (third column), a tool needs to be thin enough to reach the goal. Other attributes which affect applicability are tool length and type. In the top row, we overlay tools in yellow, which can press the buttons without colliding with the obstacles or leaving the movement space, as indicated by the green tick. The bottom row, marked with red crosses, shows tasks being attempted by orange tools, which cannot succeed without violating the constraints. Tools are rendered here for visualisation and are not present in the task images.

*tion* tasks and an *extrapolation* task. *In-sample* scenarios differ from *interpolation* scenarios in that the tasks are different but can be solved with tools (sticks or hooks) that were seen during training. The *extrapolation* scenario consists of a novel scenario but almost always also requires the use of a new tool: a J-shaped claw.

For each task, toolkit images are constructed by scattering tools from a catalogue with randomised colours and poses in a separate visible workspace. A geometric applicability check, described in Appendix B, verifies whether or not any of the tools can reach the target, while satisfying the task constraints. We consider a task–toolkit pair *feasible* if the toolkit image includes at least one tool that can achieve the task under the specified constraints.

More formally, our dataset can be written as a set of $N$ triplets: $\{(I_G^n, I_T^n, \rho^n)\}_{n=1}^N$, where each example features a task image $I_G$, a toolkit image $I_T$, and a binary label $\rho$ indicating the feasibility of reaching the target with one of the given tools (see Figure 3). We also save instance segmentation masks $M_k^n$ for the toolkits, as well as the applicability label $\rho_k^n$ for each tool, indexed by $k$. In all our experiments, we restrict the training input to the sparse high-level triplets and use the additional ground-truth labels for evaluation purposes only. For additional details on the dataset, see Appendix B.

## 4 METHOD

We frame the challenge of tool imagination as an optimisation problem in a structured latent space obtained using generative models. The optimisation is driven by a high-level, task-specific performance predictor, which assesses whether a goal position is reachable given a particular tool. To reflect real-world complexity, the system must choose a tool from a set of potentially unsuitable tools (Figure 1) to prime the optimisation. In effect, we ask how an unsuitable tool might be improved to succeed on a task. This motivates an *object-centric*, *generative* model with a latent space suited to task success classification as detailed in Figure 3.

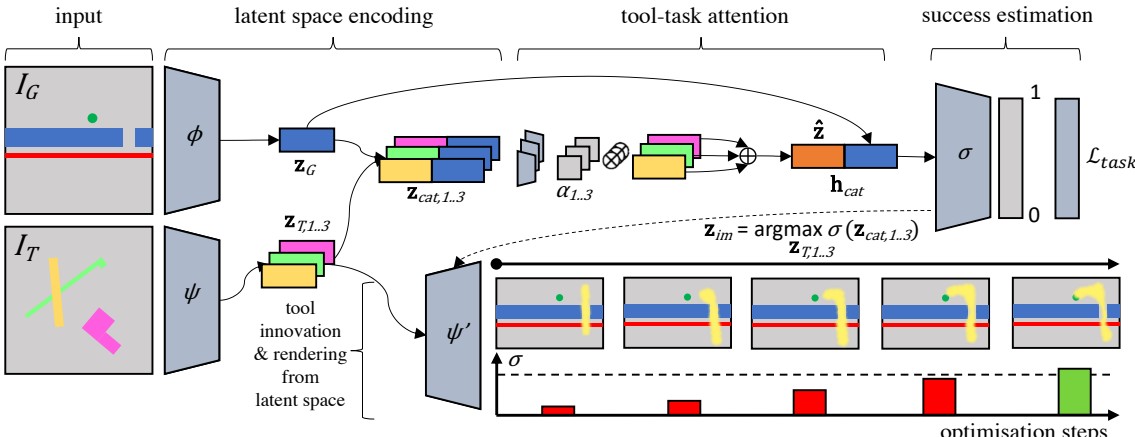

Figure 3: TasMON model architecture. A convolutional encoder $\phi$ represents the task image $I_G$ as a latent vector $\mathbf{z}_G$. In parallel, the encoder $\psi$ of a MONet scene decomposition model represents the toolkit image $I_T$ as a set of latent vectors $\mathbf{z}_{T,1..3}$ where each latent vector represents one of the tools. The concatenated tool-task representations $\mathbf{z}_{cat,1..3}$ are used to compute a soft-attention $\alpha_{1..3}$ over tools. A weighted sum $\hat{\mathbf{z}}$ of the tool representations is concatenated to the task representation again forming $\mathbf{h}_{cat}$ which is used by a classifier $\sigma$ to estimate the success of any of the tools solving the task. The task loss $\mathcal{L}_{task}$ is computed over $\sigma$'s output. We use the MONet decoder $\psi'$ to render latent tool representations into images. Given the gradient signal produced when optimising task success for the classifier $\sigma$, a chosen latent tool representation $\mathbf{z}_{im}$ gets updated to render an increasingly suitable tool for the task, as depicted on the bottom right in a callback to Figure 1. A latent vector $\mathbf{z}_4$ for the toolkit background, and a second loss $\mathcal{L}_{MONet}$ used to train the MONet model, are omitted from the figure for clarity of presentation.

## 4.1 OBJECT-CENTRIC TOOL REPRESENTATION

Given that our tools are presented in toolkit images $I_T$ (see Figure 3), it is necessary for all subsequent processing to perform a decomposition of $I_T$, from pixels into discrete objects. To achieve this decomposition of images into their component objects, we employ an unsupervised architecture called MONet (Burgess et al., 2019). MONet consists of two parts: an attention network and a component VAE (Kingma & Welling, 2014; Rezende et al., 2014). The attention network recurrently proposes $K$ regions to represent individual parts of an image. These components hopefully correspond to individual objects (plus the scene background). Each component is represented as a VAE latent code and used to explain a certain part of the input image by reconstructing it. MONet therefore serves two purposes at once: its encoder $\psi$ represents an image as a set of object-centric latent vectors, and its decoder $\psi'$ can be used to render any such latent vector into an individual object. We train MONet using the image-reconstruction loss $\mathcal{L}_{MONet}$ described in Burgess et al. (2019). We denote the set of latent variables encoding the toolkit computed by the MONet encoder, $\psi$, as

$$\psi(I_T) = \{\mathbf{z}_{T,k}\} \mid k \in \{1..K\}. \tag{1}$$

After encoding the toolkit image, we need to represent the task image $I_G$ in an abstract latent space. The task encoder, $\phi$, consists of a stack of convolutional layers followed by two dense layers.[1] $\phi$ takes a concatenation of the task image $I_G$ and a normalised meshgrid of the image's $(x,y)$ coordinates as input and maps this into the task embedding $\mathbf{z}_G$. Without loss of generality the $(x,y)$ meshgrid can be generated for any input and provides $\phi$ with a sense of relative scale via coordinate querying (see Appendix D for further details).

---

[1] Architecture details are provided in Appendix C.

In order to interpret the task success signal, we need to select the component that dominates the feasibility of the toolkit. This is done via a soft-attention approach which passes each tool representation $\mathbf{z}_{T,k}$ concatenated with the task embedding $\mathbf{z}_G$ as $\mathbf{z}_{cat,k}$ through a three-layer MLP $f$ to compute an attention logit (cf. Figure 3). A softmax activation is then applied to the logits to normalise the attention score such that

$$\alpha_k = \frac{e^{f(\mathbf{z}_{cat,k})}}{\Sigma_{j=1}^{K} e^{f(\mathbf{z}_{cat,j})}}. \tag{2}$$

The attention over $\mathbf{z}_{cat,k}$ is indicative of the influence of the $k$-th tool on global task success. We use the approximated soft attention to compute a toolkit context vector $\widehat{\mathbf{z}}$ containing information about all the tools and their respective contributions to the task solution, where

$$\widehat{\mathbf{z}} = \sum_{k=1}^{K} \alpha_k \cdot \mathbf{z}_{T,k}. \tag{3}$$

## 4.2 Tool Imagination

**Task-driven learning** The context vector $\widehat{\mathbf{z}}$ contains both task-relevant information (e.g. tool length, width, and shape), as well as task-irrelevant information such as tool location and colour. In order to perform tool imagination, the sub-manifold of the latent space that corresponds to the task-relevant features has to be found. This is achieved by adding a three-layer MLP as a classifier $\sigma$. The classifier $\sigma$ takes a concatenation $\mathbf{h}_{cat}$ of the task embedding and the context vector as input and predicts the softmax over the binary task success. The classifier learns to identify the task-relevant sub-manifold of the latent space by using the sparse success signal $\rho$ and optimising the binary-cross entropy loss, such that

$$\mathcal{L}_{task}\left(\sigma\left(\mathbf{h}_{cat}\right), \rho\right) = -\left(\rho \log\left(\sigma(\mathbf{h}_{cat})\right) + (1-\rho) \log\left(1 - \sigma(\mathbf{h}_{cat})\right)\right), \tag{4}$$

where $\rho \in \{0,1\}$ is a binary signal indicating whether or not the toolkit contains a feasible tool to solve the task. The whole system is trained end-to-end with a loss given by

$$\mathcal{L}\left(I_G, I_T, \rho\right) = \mathcal{L}_{MONet} + \lambda\mathcal{L}_{task}. \tag{5}$$

The hyper-parameter $\lambda$ denotes a weight factor balancing the task and MONet losses. Note that the gradient from the task classifier $\sigma$ propagates through both the task encoder $\phi$ and the toolkit encoder $\psi$, and therefore helps to shape the latent representations of the toolkit with respect to the requirements for task success.

**Tool imagination** Once trained, TasMON can synthesise new tools by traversing the latent manifold of individual tools in directions that maximise classification success given a toolkit image (Figure 3). To do this, we first concatenate each individual toolkit component $\mathbf{z}_k$ with the task embedding $\mathbf{z}_G$ and select the tool embedding vector $\mathbf{z}_{T,k}$ that corresponds to the highest tool utility $\sigma(\mathbf{z}_{cat,k})$ (i.e. the tool most likely to succeed). This warm-starts the imagination process. We denote the chosen tool latent as $\mathbf{z}_{im}$ and its concatenation with the task embedding vector as $\mathbf{z}_{cat,im}$. We then use activation maximisation (Erhan et al., 2009; Zeiler & Fergus, 2014; Simonyan et al., 2014) to optimise the tool encoding $\mathbf{z}_{im}$ with regard to the $\mathcal{L}_{task}$ of the success estimation $\sigma(\mathbf{z}_{cat,im})$ and a feasibility target $\rho_s = 1$, such that

$$\mathbf{z}_{im} = \mathbf{z}_{im} + \eta \frac{\partial\mathcal{L}_{task}\left(\sigma\left(\mathbf{z}_{cat,im}\right), \rho_s\right)}{\partial\mathbf{z}_{im}}, \tag{6}$$

where $\eta$ denotes the learning rate for the update. Finally, we apply this gradient update for $S$ steps or until the success estimation $\sigma(\mathbf{z}_{cat,im})$ reaches a threshold $\gamma$, and use $\psi'(\mathbf{z}_{im})$ to generate the imagined tool represented by $\mathbf{z}_{im}$.

# 5 EXPERIMENTS

Our aim in this section is to demonstrate TasMON's ability to imagine a suitable tool given a task image and an image of a putative set of tools. Concretely, we examine the model's performance when imagination requires interpolating between properties of known tool types as well as when it requires innovating new tool types that have not previously been seen during training. To provide insight into the key aspects of TasMON's operation, our main evaluation in Table 1 focuses on imagination success, while we present performance on key sub-tasks such as tool utility, decomposition, and selection as supporting work in Appendix A.

**Tool Utility (TU)**   We compute tool utility as the average precision aggregated over classifier scores $\sigma$ for all tools when compared to their ground truth feasibilities. Details are provided in Appendix A.

**Tool Imagination (TI)**   We evaluate whether our model can generate tools to achieve tasks. For each instance, the target signal $\rho$ is set to *success* such that the latent vector of the tool is then modified via back-propagation using a learning rate of $0.01$ for $5,000$ steps or until $\sigma(\mathbf{z}_{cat,im})$ reaches $0.997$. The imagined tool is then rendered via the latent space decoder $\psi'$ and tested using the same geometric applicability test as described in Appendix B. We report TI as the percentage of imagined tools that successfully pass this test.

## 5.1 QUANTITATIVE RESULTS

In order to investigate the efficacy of **TasMON** under different operating conditions, we present models trained with two different training regimes. TasMON is trained in curriculum learning manner: we first pre-train the MONet weights on toolkit images and then train the performance predictor with the MONet backbone *jointly*, i.e. the gradient from the predictor is allowed to back-propagate into the MONet. This curriculum allows MONet to learn object-centric decompositions before the performance predictor overfits the training data. To investigate the impact of jointly training the latent representation with the performance predictor we report results on an ablation model **FroMON** in which we keep the pre-trained MONet weights *frozen* during predictor training. Both TasMON and FroMON are trained on *in-sample* scenario types only. We select trained checkpoints that have the best performance prediction (TU) on a validation split. In addition to comparing the imagination outcomes for TasMON and FroMON, we also include a simple **Random**

Table 1: Like-for-like comparison of imagination processes with FroMON and TasMON, when artificially warmstarting from the same (infeasibile or feasible) tool in both cases. TI-Infeasible are the imagination results warmstarted with infeasible tools and TI-Feasible are the imagination results initialised with feasible tools.

| | | TI-Infeasible [%] | | | | TI-Feasible [%] | | |
|---|---|---|---|---|---|---|---|---|
| | n | RW | FroMON | TasMON | n | RW | FroMON | TasMON |
| Scenario | | | | | | | | |
| A | 250 | 65.2% | 87.6% | **92.0%** | 250 | 97.2% | 98.8% | **99.2%** |
| B | 250 | 25.2% | 38.0% | **40.0%** | 250 | 87.6% | **90.4%** | 84.8% |
| C | 250 | 14.4% | 26.4% | **33.2%** | 250 | 37.6% | 56.4% | **58.4%** |
| D | 250 | 8.4% | **19.6%** | 18.0% | 250 | 28.8% | 30.0% | **37.6%** |
| E | 250 | 8.4% | **20.4%** | 16.40% | 250 | **62.8%** | 61.2% | 58.8% |
| F | 250 | 6.0% | 10.8% | **13.2%** | 250 | 32.4% | **36.8%** | 30.8% |
| G | 250 | 2.0% | **5.2%** | 4.4% | 250 | 25.2% | **27.6%** | 24.4% |
| H | 250 | 0.0% | 0.0% | 0.0% | 250 | **0.4%** | **0.4%** | 0.0% |
| Overall | 2000 | 16.2% | 26.0% | **27.2%** | 2000 | 46.5% | **50.2%** | 49.3% |

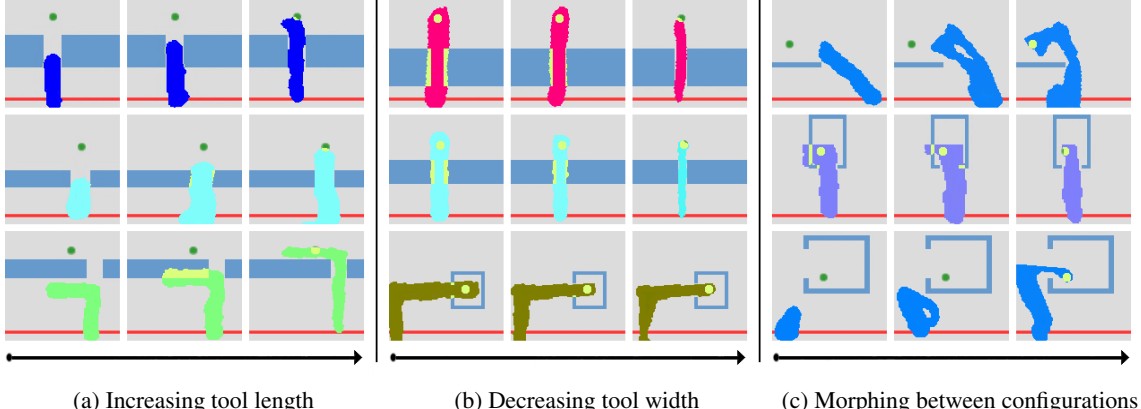

| (a) Increasing tool length | (b) Decreasing tool width | (c) Morphing between configurations |

Figure 4: Qualitative results from our imagination experiments. The figure shows the evolution of tools during the imagination process (overlayed manually over the corresponding task image). Of these, each row illustrates how the imagination procedure can succeed at constructing tools that solve a task, by (A) increasing the tool length, (B) decreasing the tool width, and (C) changing the configuration between a stick and a hook. The overlapping area is marked in bright yellow. The arrows indicates tool synthesis with increasing optimisation steps.

**Walk** (RW) baseline, where in place of taking steps in the direction of the steepest gradient, we move in a *random direction* in the latent space for 5000 steps. Specifically, the latent vector of the selected tool is updated by a sample drawn from an isotropic Gaussian with mean 0 and the absolute value of the ground-truth gradient derived by back-propagating from the predictor as the variance. For 250 infeasible instances per scenario type, we warmstart each imagination attempt with the same tool across RW, TasMON and FroMON to enable a like-for-like comparison. We also test to see how the models modify feasible tools; for 250 instances per scenario type, imagination is warmstarted with a feasible tool to observe how it is modified. The reader is referred to Table 2 and Appendix A for more quantitative results examining the model's performance in related sub-tasks like toolkit decomposition and tool selection.

TasMON successfully imagines applicable tools in 46.0% of in-sample test cases, and in 11.3% of tool interpolation test cases (Table 1). A significant drop in TI performance is incurred for the interpolation scenarios (E, F, G) in which the model encounters distinctly novel scenarios. The model fails in the extrapolation scenario where it needs to imagine a previously unseen tool type. TasMON outperforms FroMON in tool utility prediction (Table 2) and tool imagination in most tasks, suggesting that, although the predictor is powerful enough to guide the imagination through an already existing underling structure of toolkit-representations, a task-aware latent space can still provide benefits. As expected, the random walk fails in most scenarios except A and B. This is because there are no obstacles in scenario A, so any blob large enough to reach the target will suffice. We conjecture that a similar situation occurs in scenario B when the obstacle is small, which imposes only a weak constraint on the required tool. Interestingly, the imagination could also destroy some of the already feasible tools as showed in the TI-Feasible column in Table 1. Upon examination, we find that this is due to the combined effect of underconfident behaviour from the performance predictor, reconstruction errors, and rendering choices. We refer readers to Appendix E for further details about rendering.

## 5.2 QUALITATIVE RESULTS

Qualitative examples of the tool imagination process are provided in Figure 4 as well as in Figure 6 in the supplementary material. A striking feature of the optimisation process is the fact the performance predictor often drives the tools to evolve in a deliberate and interpretable way by modifying aspects such as length, width and configuration. This suggests that these properties are encoded as *directions* in the structured latent space learnt by TasMON and deliberately traversed via a high-level task objective in the form of the performance predictor. Also, since the tools are modified in a smooth manner, we hypothesise that tools are embedded in a continuous manifold of changing length, width and configuration. Moreover, instead of always turning a stick to a hook, the model can adjust the tool shape between these two configurations according to the tasks as depicted in block (c) of Figure 4, which indicates that the model learns a non-biased interpretation of the tasks.

## 6 CONCLUSION

In this paper we investigate an agent's ability to synthesise tools for simulated reaching tasks via task-driven imagination. Our approach, TasMON, uses a novel architecture in which a high-level performance predictor drives an optimisation process in a structured latent space. TasMON successfully generates tools for scenario types beyond its own training regime. Intriguingly, TasMON leads to interpretable modifications of the tools considered. The system tends to modify aspects such as tool length, width and configuration. Our results thus suggest that these object affordances are encoded as *directions* in the latent space learnt by TasMON and sought out during the optimisation process. We posit that this may help our understanding of object affordances and offers up novel avenues towards disentangling interpretable factors of variation. Nevertheless, the TasMON struggles to generalise to significantly different scenarios requiring, for example, novel tool configurations. We conjecture that this is due to the performance predictor not generalising well to these novel scenarios as the TU drops drastically in these tasks (see Table 2). Remedying this degradation in performance prediction – and thereby enabling actual tool innovation – remains an interesting challenge for future work. To facilitate further advances in this area, we intend to release the reaching dataset and trained TasMON model to the community.

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

# A  ADDITIONAL QUANTITATIVE RESULTS

In this appendix, we present performance on key sub-tasks useful for understanding model behaviour.

**Tool Utility (TU)**    The *performance predictor* $\sigma$ is responsible for estimating the probability of the toolkit containing an applicable tool for the given task based on the encoding of the task image concatenated with toolkit context vector. However, a toolkit image need not depict more than a single tool, and so $\sigma$ can provide a measure of tool utility of an individual, possibly imagined, tool. We use intersection-over-union (IOU) as a similarity measure to match tool decompositions to ground truth tool masks according to the Kuhn-Munkres algorithm (Kuhn, 1955; Munkres, 1957). All matches below the conservative threshold of IOU $< 0.8$ are discarded as they correspond to spurious or poor quality decompositions. Matched ground-truth masks are assigned their corresponding tool decomposition's tool utility probability from the classifier. We report the aggregated tool utility as average precision. A tool utility probability of 0 is assigned to any unmatched ground truth masks. Since we report average precision, this rubric can only deflate our scores by increasing false negatives at the cost of true positives, leading to conservative estimates of performance.

**Toolkit Decomposition (TD)**    We use the *Adjusted Rand Index (ARI)* (Rand, 1971; Hubert & Arabie, 1985) to evaluate the quality of the instance segmentation, as in Greff et al. (2019). An ARI score treats the pixel-to-instance-assignment as a clustering problem and computes the similarity between the predicted clustering and ground truth. ARI scores range from 0 (complete misalignment) to 1 (identical clustering).

**Tool Selection (TS)**    The TasMON model performs tool selection as described in Section 4.2. In feasible scenarios, we select the tool decomposition with the highest tool utility, and compare it with the ground truth mask of the applicable tool. We consider tool selection to be successful if the predicted mask has an IOU overlap with the ground-truth mask of at least 0.8, which we take to be a conservative threshold.

Table 2: Results table, comparing TasMON with a frozen MONet+Task ablation, i.e. FroMON, across all tasks. Other abbreviations: TU = tool utility, aggregated as the average-precision score over all ground truth tools; TD = toolkit decomposition, measured by the Adjusted Rand Index (Rand, 1971; Hubert & Arabie, 1985); TS = tool-selection success in feasible scenarios, % of times pixel-wise IOU of most-likely tool and the ground-truth feasible tool was 0.8 or more; TI = unconditional tool-imagination success for each model, % of all test scenarios; The best model for a task and metric are shown in **bold**, except for TD where ARI scores were similar for different models. All metric scores are shown as percentages for consistency. Like-for-like tool-imagination results are shown in Table 1. Please refer to Figure 2 for illustrative tasks by scenario.

| | TU [%] | | TD [%] | | TS [%] | | TI [%] | |
|---|---|---|---|---|---|---|---|---|
| | FroMON | TasMON | FroMON | TasMON | FroMON | TasMON | FroMON | TasMON |
| Scenario | | | | | | | | |
| A | 95.4% | **97.6%** | 97.7% | 97.7% | 95.2% | **96.2%** | 83.9% | **87.4%** |
| B | 88.9% | **91.3%** | 97.8% | 97.8% | **82.8%** | 79.8% | 48.6% | **48.9%** |
| C | 83.2% | **85.8%** | 97.7% | 97.9% | 88.0% | **88.2%** | 38.2% | **40.8%** |
| D | 77.2% | **79.4%** | 97.8% | 97.8% | **78.6%** | 75.8% | 20.2% | 20.2% |
| E | 76.3% | **77.3%** | 97.7% | 97.8% | **72.0%** | 71.4% | **29.1%** | 24.9% |
| F | **37.9%** | 36.5% | 97.7% | 97.9% | 62.4% | **65.2%** | 15.8% | **16.0%** |
| G | **48.6%** | 48.4% | 97.8% | 97.8% | **76.6%** | 73.6% | **14.2%** | 12.4% |
| H | 37.8% | **38.4%** | 96.9% | 97.1% | 69.6% | **70.0%** | **0.1%** | 0.0% |
| Overall | 68.2% | **69.3%** | 97.6% | 97.7% | **78.2%** | 77.5% | 31.3% | 31.3% |

Table 2 presents model performance across all four metrics. Note that the tool-imagination (TI) results here differ from Table 1 because in this table, we do not warmstart imagination with the same tool for each model, instead letting each model select a tool with highest TU. As a result, models might select different tools to bootstrap imagination, leading to a comparison that is not exactly like-for-like.

We see that both models demonstrate a strong ability to gauge tool utility (Table 2, TU column) in *in-sample* scenarios. However, prediction performance decreases as scenario types grow more distinct from the training data. Table 2 suggest a strong correlation between successful tool imagination (TI) and effective and robust tool utility prediction (TU). This is unsurprising as the performance predictor constitutes the main driver for the optimisation. Finally, we note that task decomposition (TD) is commensurate across all models and scenario types. This is an intuitive result as both models are trained with an explicit decomposition objective, which is independent of scenario type.

## B  DATASET CONSTRUCTION

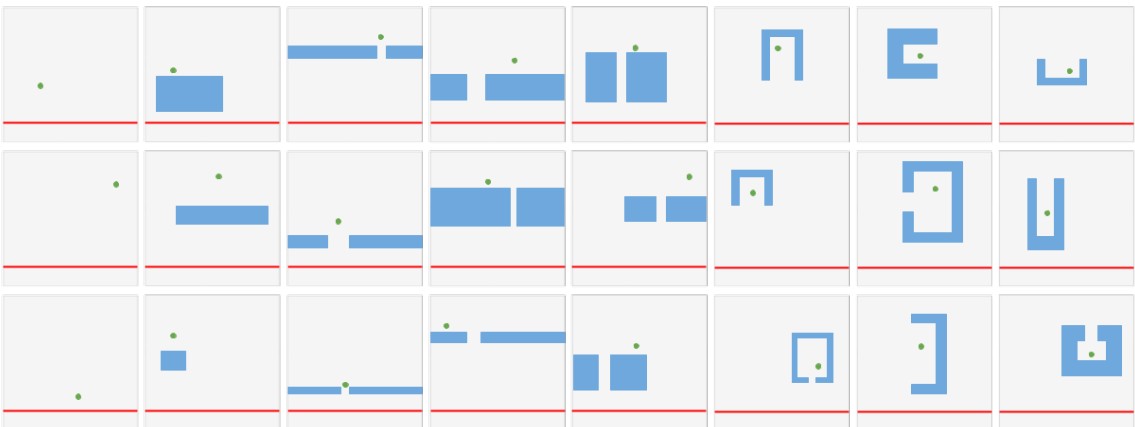

Figure 5: More illustrative examples of tasks (cf. Figure 2). We procedurally generated 196,390 unique instances across eight scenario types for our reaching experiments.

Our synthetic dataset consists of pairs of tasks and toolkits rendered from top-down views as $64 \times 64$ pixel RGB images. We create the dataset in multiple sampling steps ensuring a balance between scenarios and tool configurations. For each scenario, we generate 5,000 tasks. Each is paired with 50 tools from a tool catalogue containing 1 million unique tool geometries, evenly balanced between sticks, hooks and claw tools. Then we perform the tool applicability check for each of the task-tool pairs and reject tasks which had fewer than five or more than 40 applicable tools. The applicability of a tool to a task is determined by sampling 200 interior points of the tool polygon, overlaying the sampled point with the target and rotating the tool polygon and a vertically mirrored copy of it by 360 degrees. If any such pose of the tool satisfies all constraints (i.e. touching the space behind the red line while not colliding with any obstacle), we consider the tool applicable for the given task. In this way, we construct up to five feasible and five infeasible toolkits for each task, by sampling one to three tools per toolkit and rendering them with randomised poses and colours. We constrain the feasible toolkits to contain *exactly one* applicable tool. All toolkits are constrained to have only sticks and hooks as tools, with the exception of the H tasks, which require claws in order to be feasible.

The dataset contains a total of 52,000 scenarios. Table 3 shows a breakdown of each by scenario type and split. Each split has an equal number of feasible and infeasible instances, for each scenario type.

Table 3: Number of instances by scenario type.

| Type | Training | Validation | Test |
|------|----------|------------|------|
| A | 10,000 | 1,000 | 1,000 |
| B | 10,000 | 1,000 | 1,000 |
| C | 10,000 | 1,000 | 1,000 |
| D | 10,000 | 1,000 | 1,000 |
| E | - | - | 1,000 |
| F | - | - | 1,000 |
| G | - | - | 1,000 |
| H | - | - | 1,000 |
| Total | 40,000 | 4,000 | 8,000 |

## C  ARCHITECTURE AND TRAINING DETAILS

### C.1  MODEL ARCHITECTURE

As described in the main text, our model (*TasMON*) comprises two main parts (as depicted in Figure 3). On the one hand, we used a task-based classifier (*Tas*), while on the other we used a MONet as a scene decomposition model (*MON*). We briefly describe each component in turn.

#### C.1.1  MONET

As a scene decomposition model, we faithfully re-implement MONet (Burgess et al., 2019) with the minor modification of using leaky ReLUs (Maas et al., 2013) as activation functions in place of ReLUs.

#### C.1.2  TASK-BASED CLASSIFIER

The task-based classifier consists of three sub-parts: a task encoder, an attention module, and a classifier.

The task encoder stacks three convolutional blocks followed by a single convolutional layer and a two-layer MLP. First, the three consecutive convolutional blocks make use of 16, 32, 64 output channels respectively. Each convolutional block contains two consecutive sub-convolutional blocks, each with a kernel size of 3 and padding of 1 (and with stride of 2 and stride of 1, respectively). Second, the additional convolutional layer has stride 2, padding 1, and 128 output channels. All convolutional layers use a leaky ReLU nonlinearity. Finally, the MLP takes a flattend output of the convolutional layers of dimension 2048 ($4 \times 4 \times 128$) and passes it to a hidden layer with 128 neurons using a leaky ReLU as the activation function. This is then passed through another layer, giving a 16 dimensional output for the task-based classifier.

The attention module is a three layer MLP with input dimension 32, and successive hidden layers of 128 and 16 neurons respectively. Each of these layers use leaky ReLU activation functions. To predict the logit of the attention probability, the attention module has an output dimension of 1.

The structure of the classifier part of the task-classifier model mirrors the attention module. In place of the output layer, it consists of a three layer MLP with the same dimensions. Given that the classifier outputs logits for a binary classifier, the output dimension is 2.

### C.2  TRAINING DETAILS

All experiments are performed in PyTorch, using the ADAM optimiser with a learning rate of 0.0001 and a batch size of 32. We vary the weight factor $\lambda$ between 0.1 to 4, and we select TasMON to be the best performing model, at $\lambda = 1$. MONet's hyperparameters tuned to our dataset are given in Table 4.

Table 4: Hyperaparmeters used for pretrained MONet.

| MONet Hyperparameters | |
| --- | --- |
| Attention Blocks | 5 |
| Attention Channels | 8 |
| Beta | 0.1 |
| Gamma | 0.3 |
| Pixel variance background | 0.18 |
| Pixel variance foreground | 0.2 |
| VAE Channels | 32 |
| VAE Latents | 16 |

## D    COORDINATE QUERYING

The CNN is local and its parameters are shared across spatial locations; however, the location of an object in an image is global information. In order to infer the global location of an object, the CNN must work together with the permutation-variant dense-layers. Assume that in a CNN-Dense network structure the output shape of the feature tensor just before the flatten operation is $3 \times 3$ and we only take the middle anchor point feature, i.e. the one located at$(1, 1)$, whose receptive field does not touch the boundaries of the task image. Based only on that feature vector the network will not be capable to predict the global coordinates of the object observed within that receptive filed. The global coordinates of an object can only be inferred by analysing the information of a subset of all the anchor point features that have a joint receptive field larger than or equal to the whole image. This will also utilise some capacity of the dense layers. But given the x-y meshgrid attached to the task image, even the kernel in the input layer with a receptive field of 3 (assuming a kernel size of $3 \times 3$) can know where it is, which allows the network to query the global coordinate information of an object locally. In conclusion, the x-y meshgrid is free additional information that can help the task module to capture the relative scale of the scene.

## E    RENDERING METHOD

Although the MONet aims to decompose a scene into object-centric representations, the imagined component cannot be rendered independently from the rest parts of the image. This is because the reconstruction outputs of the decoder are the unnormalised logits and directly using softmax to normalise the mask of the imagined component together with the rest parts will constrain the growth of the tool. Imagine that the imagination process that tries to enlarge a tool is assigning higher logits to the area where it thinks the tool should grow towards, while the logits of that area of the background component are still high since the imagination process is not applied to it. During normalisation the imagined component must beat the background to possess the pixels it wants to grow towards and thus the growth is constrained implicitly. We tackle this problem by rather comparing the logits of the imagined tool with the rest components but comparing with itself. We use the minimum logit of the area where the attention mask has a (normalised) probability higher than 0.2 as a threshold before the imagination starts. Any imagined pixel that has a logit higher than that threshold is treated as belonging to the imagined tool. It's common that when people develop a scene decomposition model they normalise the component masks altogether in the end since "sum up to one" is an important inductive bias by design. Nevertheless, our work indicates that this might constrain the down-streaming tasks such as optimising the object-centric representation independently for planning purpose.

## F ADDITIONAL IMAGINATION EXAMPLES

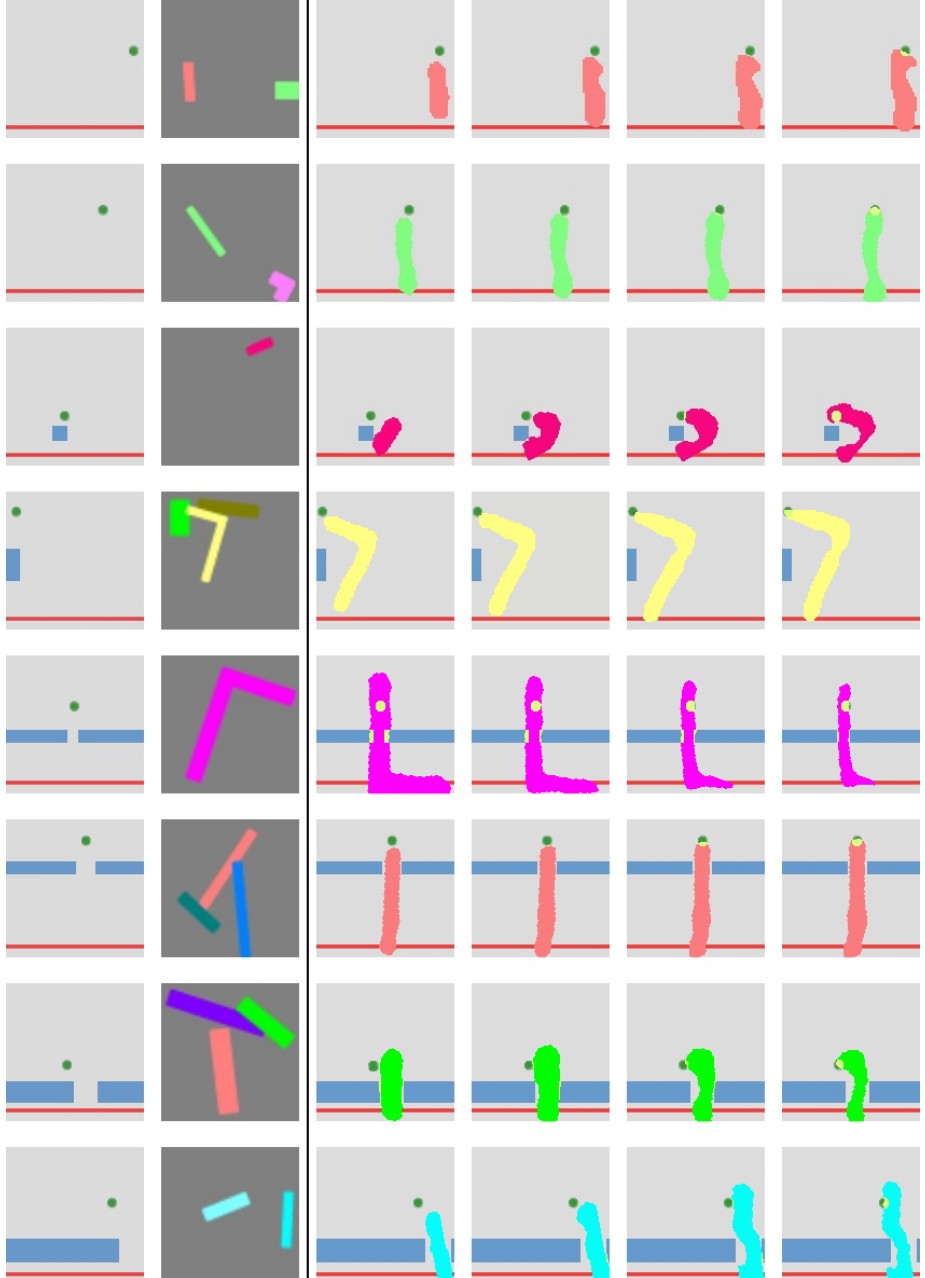

Figure 6

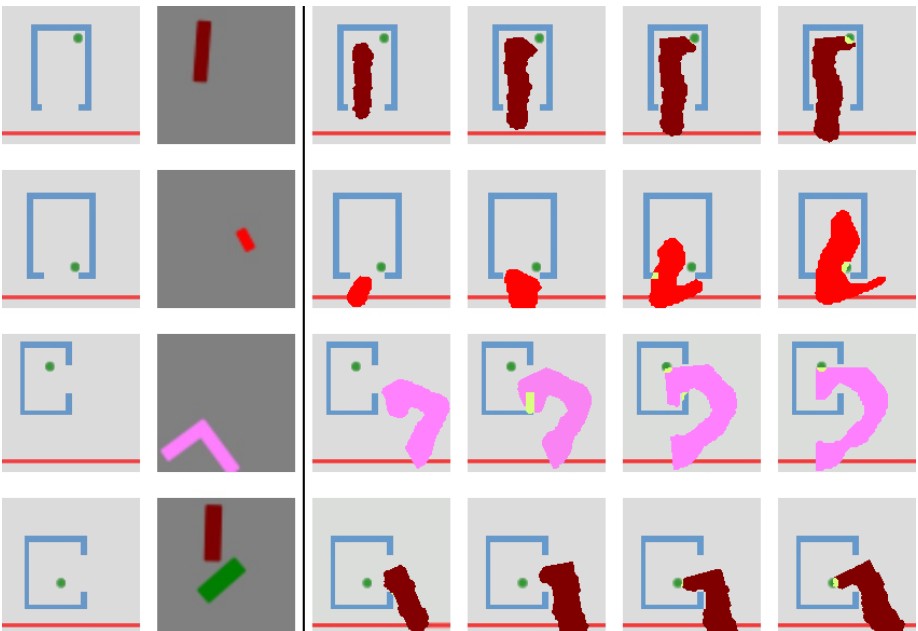

Figure 6: More examples of imagined tools for all tasks. The first two columns are the task and toolkit inputs, and the next four columns are imagination reconstructions for the selected tool overlaid on the task image. The overlapping area is marked in bright yellow.

