# OpenReview forum: "Imagine That! Leveraging Emergent Affordances for Tool Synthesis in Reaching Tasks"
_ICLR.cc/2020/Conference — Reject_

### Official Review · AnonReviewer2 · 2019-10-18
**Official Blind Review #2**

**Rating:** 3

**Review:**

The authors constructed an interesting dataset named reaching task, where the model need to predict if the given toolkit is able to solve the corresponding task or not. They showed that combining variational auto-encoding with an auxiliary loss (in this case, the predictor of solving the tasks, could help shaping a latent space where affordance of the tool is encoded as directions in such latent space.) Using the activation maximisation technique (which was phrased as the imagination process), they are able to modify the input tools into the ones that is suitable to solve the corresponding task. I found the idea of using an auxiliary loss when training a VAE may cause the latent space coding direction change novel and interesting. However, I do not find the authors has a strong case of proven it is the case in this manuscript.

1. The performance difference between FroMON and TasMON is not clear.
  The most critical control model in this paper is the FroMON (frozen MoNet). In this control model, the gradient from the success predictor is not flowing back into the VAE encoder. So, based on the author's assumption, it should not be benefit of having the tool affordance directions in the latent space. However, in the main results in Table 1. We found the performance between FroMON and TasMON is not quite clear. This is particularly true for the Scenario E, F, G (the interpolation tasks), which is more about generalization and is more important.

2. Are the affordance 'directions' in the latent space?
  The authors used activation maximisation approach to travel in the latent space. My understanding of the approach is it follow the gradient to maximise the predictor's success prediction in an iterative approach. So, at each optimization step, the z_im can move in different direction. This seems to not fit as a sense of 'direction', as I would assume it is moving along a particular line (not necessarily axis aligned.). Maybe this does explain whey FroMON and TasMon perform equally well. As long as the possible shapes is encoded in a smooth way in the latent space, the activation maximisation could find a path toward the target object. Unfortunately, is that a 'direction'? Would it be possible to train an optimization algorithm that is only allow to move in a linear direction, and see how well that work?

3. Why MoNet and multiple tools in the toolkit. A simplified version could drive the point as well.
Using MoNet to decompose tools from a toolkit is nice. However, is it really necessary to drive the main point (an auxillary loss of success prediction can shape the latent space of a VAE model) in this paper. In a simplified version, where there is only one tool in the toolkit, one may not need MoNet (maybe still need it for object-background separation?) May the authors comment why multiple tools in the toolkit is important?

Minor:
1. typo: page 1, (2nd to the last line). '...that habitual tool use cannot in and off itself ..' --> of
2. A simple video showing how the tool shape change sequentially during the activation maximisation process would be interesting.

**Experience Assessment:**

I have read many papers in this area.

**Review Assessment: Checking Correctness Of Derivations And Theory:**

I assessed the sensibility of the derivations and theory.

**Review Assessment: Checking Correctness Of Experiments:**

I carefully checked the experiments.

**Review Assessment: Thoroughness In Paper Reading:**

I read the paper at least twice and used my best judgement in assessing the paper.

---

> ### Author Response · Authors · 2019-11-12
> **Response to Reviewer 2**
>
> We would like to thank Reviewer 2 for their review and constructive suggestions. Our responses:
>
> Q: The performance difference between FroMON and TasMON is not clear. The most critical control model in this paper is the FroMON (frozen MoNet). In this control model, the gradient from the success predictor is not flowing back into the VAE encoder. So, based on the author's assumption, it should not be benefit of having the tool affordance directions in the latent space. However, in the main results in Table 1. We found the performance between FroMON and TasMON is not quite clear. This is particularly true for the Scenario E, F, G (the interpolation tasks), which is more about generalization and is more important.
>
> A: This is a misunderstanding. Please see our overall comment 3) above. We mention “a task-aware latent space can still provide benefits” because the tool-utility (TU) shown in Table 2 get improved in most tasks. And the metric TU captures how well does the classifier understand the tool utility given a learned latent space. The performance difference between FroMON and TasMON in terms of TI is not clear because the imagination performance is not only determined by the quality of the latent space but also by many other effects such as the rendering methods, the hyper-parameter of imagination steps as explained in the appendix E. Therefore, we emphasise the fact that the tool-imagination can still work with FroMON by saying “although the predictor is powerful enough to guide the imagination through an already existing underlying structure of toolkit-representations”. The distinction between FroMON and TasMON is thus tangential to the main thrust of our work, which is that task-relevant object affordances are implicitly encoded as trajectories in the latent space and these can be leveraged by traversing along a trajectory driven by a high-level performance predictor.
>
> Q: Are the affordance 'directions' in the latent space? The authors used activation maximisation approach to travel in the latent space. My understanding of the approach is it follows the gradient to maximise the predictor's success prediction in an iterative approach. So, at each optimization step, the z_im can move in different directions. This seems to not fit as a sense of 'direction', as I would assume it is moving along a particular line (not necessarily axis aligned.). Maybe this does explain whey FroMON and TasMon perform equally well. As long as the possible shapes is encoded in a smooth way in the latent space, the activation maximisation could find a trajectory toward the target object. Unfortunately, is that a 'direction'? Would it be possible to train an optimization algorithm that is only allow to move in a linear direction, and see how well that work?
>
> A: Please see our overall comment 4) above. We concede that our use of the phrase ‘directions in latent space’ would be better expressed as ‘local directions in latent space’, or ‘trajectories in latent space’. The important point, however, is that these trajectories link to the semantic of object affordances.
>
> Q: Why MoNet and multiple tools in the toolkit. A simplified version could drive the point as well. Using MoNet to decompose tools from a toolkit is nice. However, is it really necessary to drive the main point (an auxillary loss of success prediction can shape the latent space of a VAE model) in this paper. In a simplified version, where there is only one tool in the toolkit, one may not need MoNet (maybe still need it for object-background separation?) May the authors comment why multiple tools in the toolkit is important?
>
> A: This is a misunderstanding. The main point of our paper is in fact task relevant object affordances are implicitly encoded as [trajectories] in a structured latent space shaped by experience and that we can access them with optimisation of the latent encoding via a high-level performance predictor. Our results from Tables 1 and 2 indicate that TasMON and FroMON have similar performance. We feel that the following statement in our paper might have caused this misunderstanding: “TasMON outperforms FroMON in tool utility prediction (Table 2) and tool imagination in most tasks, suggesting that, although the predictor is powerful enough to guide the imagination through an already existing underlying structure of toolkit-representations, a task-aware latent space can still provide benefits.” We will clarify this in the next version. Thank you for helping us to refine our presentation.
>
> We also thank the reviewer for pointing out the typo; we have corrected it.

---

> > ### Comment · AnonReviewer2 · 2019-11-15
> > **Thank you for a detailed reply to the comments.**
> >
> > Thank you for your detailed explanations to the concerns.
> > I think this is certainly an interesting topic. However, I still think the results demonstrated in the current work is not strong enough to convince people. So, I would stick with the current score.

---

### Official Review · AnonReviewer3 · 2019-10-22
**Official Blind Review #3**

**Rating:** 1

**Review:**

This paper proposes an architecture for synthesizing tools to be used in a reaching task. Specifically, during training the agent jointly learns to segment an image of a set of three tools (via the MONet architecture) and to classify whether one the tools will solve the given scene. At test time, one of the three tools is selected based on which seems most feasible, and then gradient descent is used to modify the latent representation of the tool in order to synthesize a new tool to (hopefully) solve the scene. The paper demonstrates that this approach can achieve ok performance on familiar scenes with familiar tools, but that it fails to generalize when exposed to unfamiliar scenes or unfamiliar tools. The paper reports a combination of the quantitative results showing that optimizing the latent space can lead to successful synthesis in some cases, and qualitative results showing that the synthesized tools change along interpretable dimensions such as length, width, etc. The combination of these results suggest that the model has learned something about which tool dimensions are important for being able to solve the types of reaching tasks given in the paper.

While I think this paper tackles a very interesting, important, and challenging problem, I unfortunately feel it is not ready for publication at ICLR and thus recommend rejection. Specifically, (1) neither the particular task, results, or model are not very compelling, (2) there are no comparisons to meaningful alternatives, and (3) overall I am not quite sure what conclusions I should draw from the paper. However, given the coolness of the problem of tool synthesis, I definitely encourage the authors to continue working on this line of work!

1. The task, results, and model are not very compelling. Any of these three things alone would not necessarily be a problem, but given that all three are true the paper comes across as a bit underwhelming.

- First, while the task can be construed as a tool synthesis task, it doesn’t come across to me as very ecologically valid. In fact, the task seems to be more like a navigation task than a tool synthesis task: what’s required is simply to draw an unbroken line from one part of the scene to another, rather than actually generate a tool that has to be manipulated in an interesting way. Navigation has been studied extensively, while synthesis of tools that can be manipulated has not, which makes this task both not very novel and disappointing in comparison to what more ecologically-valid tool synthesis would look like. For example, consider a variation of the task where you would have to start the tool at the red region and move it to the green region. Many of the tools used here would become invalid since you wouldn’t actually be able to fit them through the gaps (e.g. Figure 2E).

- Second, given that the “synthesis” task is more like a navigation task, the results are somewhat disappointing. When provided with a feasible solution, the model actually gets *worse* even in some of the in-sample scenes that it has seen during training (e.g. scene types C and D) which suggests that it hasn’t actually learned a good generative model of tools. Generalization performance is pretty bad across the board and is only slightly better than random, which undermines the claim in the abstract that “Our experiments demonstrate that the synthesis process modifies emergent, task-relevant object affordances in a targeted and deliberate way”. While it’s clear there is successful synthesis in some cases, I am not sure that the results support the claim that the synthesis is “targeted” or “deliberate” given how poor the overall performance is.

- Third, the model/architecture is a relatively straightforward combination of existing components and is highly specialized to the particular task. As mentioned above, this wouldn’t necessarily be a problem if the task were more interesting (i.e. not just a navigation task) and if the results were better. I do think it is cool to see this use of MONet but I’m skeptical that the particular method of optimizing in the latent space is doing anything meaningful. While there is prior work that has optimized the latent space to achieve certain tasks (as is cited in the paper), there is also a large body of work on adversarial examples which demonstrate that optimizing in the latent space is also fraught with difficulty. I also suspect this is the reason why the results are not particularly good.

2. While I do appreciate the comparisons that are in the paper (to a “Random” version of TasMON that moves in a random direction in the latent space, and to “FroMON” agent which is not allowed to backpropagate gradients from the classification loss into MONet), these comparisons are not particularly meaningful. The difference between FroMON performance and TasMON tool imagination performance (I didn’t test tool utility) across tasks is not statistically significant (z(520, 544)=-0.8588, p=.38978), so I don’t think it is valid to claim that “a task-aware latent space can still provide benefits.” The Random baseline is a pretty weak baseline and it would be more interesting to compare to an alternative plausible architecture (for example, which doesn’t use a structured latent space, or which doesn’t have a perceptual frontend and operates directly on a symbolic representation of the tools/scene).

3. Overall, I am not quite sure what I am supposed to get out of the paper. Is it that “task relevant object affordances are implicitly encoded as directions in a structured latent space shaped by experience”? If so, then the results do not support this claim and so I am not sure what to take away. Is it that the latent space encodes information about what makes a tool feasible? If so, then this is a bit of a weak argument---of *course* it must encode this information if it is able to do the classification task at all. Is it that tool synthesis is a challenging problem? If so, then the lack of strong or canonical baselines makes it hard to evaluate whether this is true (and the navigation-only synthesis task also undermines this a bit).

Some additional suggestions:

It would be good to include a discussion of other recent work on tool use such as Allen et al. (2019) and Baker et al. (2019), as well as on other related synthesis tasks such as Ha (2018) or Ganin et al. (2018).

The introduction states that “tool selection and manufacture – especially once demonstrated – is a significantly easier task than tool innovation”. While this may be true, it is a bit misleading in the context of the paper as the agent is doing something more like tool selection and modification rather than tool innovation (and actually the in-sample scenes are more like “manufacture”, which the agent doesn’t always even do well on).

It would be helpful to more clearly explain scene types. Here is some suggested phrasings: in-sample = familiar scenes with familiar tools, interpolation = novel scenes with familiar tools, extrapolation = novel scenes with novel tools.

I was originally confused how psi’ knew where to actually place the tool and at what orientation, and whether the background part of the rendering process shown in Figure 1. I realized after reading the supplemental that this is not done by the agent itself but by separate code that tries to find the orientation and position of the tool. This should be explained more clearly in the main text.

In Table 1 it would be helpful to indicate which scene types are which (in-sample, interpolation, extrapolation).

Allen, K. R., Smith, K. A., & Tenenbaum, J. B. (2019). The Tools Challenge: Rapid Trial-and-Error Learning in Physical Problem Solving. arXiv preprint arXiv:1907.09620.

Baker, B., Kanitscheider, I., Markov, T., Wu, Y., Powell, G., McGrew, B., & Mordatch, I. (2019). Emergent tool use from multi-agent autocurricula. arXiv preprint arXiv:1909.07528.

Ganin, Y., Kulkarni, T., Babuschkin, I., Eslami, S. M., & Vinyals, O. (2018). Synthesizing programs for images using reinforced adversarial learning. arXiv preprint arXiv:1804.01118.

Ha, D. (2018). Reinforcement learning for improving agent design. arXiv preprint arXiv:1810.03779.

**Experience Assessment:**

I have published one or two papers in this area.

**Review Assessment: Checking Correctness Of Derivations And Theory:**

N/A

**Review Assessment: Checking Correctness Of Experiments:**

I assessed the sensibility of the experiments.

**Review Assessment: Thoroughness In Paper Reading:**

I read the paper at least twice and used my best judgement in assessing the paper.

---

> ### Author Response · Authors · 2019-11-12
> **Response to Reviewer 3**
>
> We would like to thank Reviewer 3 for their review and constructive suggestions. Our responses:
>
> Q: First, while the task can be construed as a tool synthesis task, it doesn’t come across to me as very ecologically valid. In fact, the task seems to be more like a navigation task than a tool synthesis task: what’s required is simply to draw an unbroken line from one part of the scene to another, rather than actually generate a tool that has to be manipulated in an interesting way. Navigation has been studied extensively, while synthesis of tools that can be manipulated has not, which makes this task both not very novel and disappointing in comparison to what more ecologically-valid tool synthesis would look like. For example, consider a variation of the task where you would have to start the tool at the red region and move it to the green region. Many of the tools used here would become invalid since you wouldn’t actually be able to fit them through the gaps (e.g. Figure 2E).
>
> A: While our work is firmly rooted in the literature on tool use (e.g. [1][2][3]) we agree that our problem setup is also reminiscent of planning and navigation tasks. We see this as an opportunity to apply our approach to these fields rather than an indication that our work is framed in an inappropriate context. Also, our work shows that the appearance of an object can be planned. The task-relevant variations can be captured precisely and manipulated deliberately via a performance predictor.
>
> Please see our discussion 1) about the explanation of valid tool.
>
> We are unaware of other papers using our approach, exploiting a performance predictor to optimise a latent code and generate potential solutions having been explored in these domains. We would be grateful if the reviewer could point us to such work so we can incorporate it into our related work section.
>
> We note that the dataset is designed to contain three task-relevant variations (affordances), e.g. length, width, shape (hook-length) and some other task-irrelevant variations, e.g. colour and location of the tool. The model is expected to capture the task-relevant variations and neglect the irrelevant ones.  Moreover, given a specific task that exposes constraints on one particular or a combination of affordances, the model should be able to not only understand which kind of affordance needs to modified but also guide the traversal in the latent space along a trajectory. Traversing a “trajectory” (what we previously called a “direction”) in the latent space corresponds exactly to the modification of one kind of affordance in the image space, as depicted in Fig 4. To the best of our knowledge, we are the first to link the concept of affordance in this way to following trajectories in latent space.
>
> [1] Nathan J. Emery and Nicola S. Clayton. Tool use and physical cognition in birds and mammals. Current Opinion in Neurobiology, 19(1):27 – 33, 2009.
>
> [2] Jackie Chappell and Alex Kacelnik. Tool selectivity in a non-primate, the New Caledonian crow (Corvus moneduloides). Animal Cognition, 5(2):71–78, 2002.
>
> [3] Jackie Chappell and Alex Kacelnik. Selection of tool diameter by New Caledonian crows Corvus moneduloides. Animal Cognition, 7(2):121–127, 2004.

---

> ### Author Response · Authors · 2019-11-12
> **Response to Reviewer 3**
>
> Q: Second, given that the “synthesis” task is more like a navigation task, the results are somewhat disappointing. When provided with a feasible solution, the model actually gets *worse* even in some of the in-sample scenes that it has seen during training (e.g. scene types C and D) which suggests that it hasn’t actually learned a good generative model of tools. Generalization performance is pretty bad across the board and is only slightly better than random, which undermines the claim in the abstract that “Our experiments demonstrate that the synthesis process modifies emergent, task-relevant object affordances in a targeted and deliberate way”. While it’s clear there is successful synthesis in some cases, I am not sure that the results support the claim that the synthesis is “targeted” or “deliberate” given how poor the overall performance is.
>
> A: We agree that if we treat the tasks as navigation tasks then the tasks will be naive and several path planning algorithms can tackle them. But “these path planning problems are solvable” is not the point we want to make. We design this dataset such that it controls explicitly three kinds of task-relevant variations (length, width, shape/hook-length) and other task-irrelevant variations (colour, location). The model is expected to capture and identify the task-relevant ones only given weak task success/failure singal. The captured task-relevant variations are then used for tool-selection (Table 2 in the appendix), and tool-imagination (figure 4). This simplified design is aimed at emulating a recent finding [1] in biological science, which shows that crows not only use tools but also improvise better ones to reach the food in a puzzle box.
>
> [1] Bayern, A.M.P.v., Danel, S., Auersperg, A.M.I. et al. Compound tool construction by New Caledonian crows. Sci Rep 8, 15676 (2018) doi:10.1038/s41598-018-33458-z
>
> Q: Third, the model/architecture is a relatively straightforward combination of existing components and is highly specialized to the particular task. As mentioned above, this wouldn’t necessarily be a problem if the task were more interesting (i.e. not just a navigation task) and if the results were better. I do think it is cool to see this use of MONet but I’m skeptical that the particular method of optimizing in the latent space is doing anything meaningful. While there is prior work that has optimized the latent space to achieve certain tasks (as is cited in the paper), there is also a large body of work on adversarial examples which demonstrate that optimizing in the latent space is also fraught with difficulty. I also suspect this is the reason why the results are not particularly good.
>
> A: We agree that we leverage known mechanisms but would argue that this does not contradict the contribution of our architecture design (which traverses the latent space using a high-level description of the tasks and the conditional activation maximisation). We might further argue that it is a strength of our work that it points to the potential that the traditional planning tasks in robotics can be cast into a problem of modifying the task-relevant variations represented by the latent embedding using high-level task-predictors. Even the appearance of an object can be planned.
>
> Q: While I do appreciate the comparisons that are in the paper (to a “Random” version of TasMON that moves in a random direction in the latent space, and to “FroMON” agent which is not allowed to backpropagate gradients from the classification loss into MONet), these comparisons are not particularly meaningful. The difference between FroMON performance and TasMON tool imagination performance (I didn’t test tool utility) across tasks is not statistically significant (z(520, 544)=-0.8588, p=.38978), so I don’t think it is valid to claim that “a task-aware latent space can still provide benefits.” The Random baseline is a pretty weak baseline and it would be more interesting to compare to an alternative plausible architecture (for example, which doesn’t use a structured latent space, or which doesn’t have a perceptual frontend and operates directly on a symbolic representation of the tools/scene).
>
> A: Thank you for this insightful comment. At the time, we aimed to keep comparisons limited to ablations in order to verify the efficacy of the proposed architecture and to avoid confounders. A solution that uses ground-truth symbolic/physical representations of objects and tasks would be a good upper-bound baseline. We note that the Pix2Pix model can also be used to generate realistic feasible tools if we synthesis the corresponding feasible tools as additional supervision although it can not turn an infeasible tool to a feasible one. We will evaluate these in the next iteration.

---

> ### Author Response · Authors · 2019-11-12
> **Response to Reviewer 3**
>
> Q: Overall, I am not quite sure what I am supposed to get out of the paper. Is it that “task relevant object affordances are implicitly encoded as directions in a structured latent space shaped by experience”? If so, then the results do not support this claim and so I am not sure what to take away. Is it that the latent space encodes information about what makes a tool feasible? If so, then this is a bit of a weak argument---of *course* it must encode this information if it is able to do the classification task at all. Is it that tool synthesis is a challenging problem? If so, then the lack of strong or canonical baselines makes it hard to evaluate whether this is true (and the navigation-only synthesis task also undermines this a bit).
>
> A: Yes, “task relevant object affordances are implicitly encoded as directions/trajectories in a structured latent space shaped by experience”. However, we should add that these directions need not be standard basis vectors (i.e. axis-aligned) with respect to the coordinate system of the latent space for our approach of using activation maximisation to work. Our experiments show that perturbing the latent representation of a tool in the direction, revealed by conditional activation-maximization, results in modifying the tool’s semantic task-relevant affordances, which supports the above claim. Other aspects of the tool such as its colour, or its position in the visual field, which are orthogonal to solving tasks, remain interestingly unchanged. That this precise perturbation of object properties can be achieved by a high-level performance predictor is, we believe, interesting and of great potential use to a variety of other applications.
>
> S: It would be good to include a discussion of other recent work on tool use such as Allen et al. (2019) and Baker et al. (2019), as well as on other related synthesis tasks such as Ha (2018) or Ganin et al. (2018).
>
> A: We thank the reviewer  for introducing these interesting papers. We will add these into the related work section in the next iteration.
>
> S: I was originally confused how psi’ knew where to actually place the tool and at what orientation, and whether the background part of the rendering process shown in Figure 1. I realized after reading the supplemental that this is not done by the agent itself but by separate code that tries to find the orientation and position of the tool. This should be explained more clearly in the main text.
>
> A: Please refer to general comment 1). It is not the task of the agent to put the tool in the correct pose. The agent only learns to modify the shape of the tool. The tool and task is put together manually for illustration of the success of the imagination. However, the success check is done by the geometric applicability check introduced in the  Appendix B. We will explain this more clearly in the main text in the next iteration.
>
> S: The introduction states that “tool selection and manufacture – especially once demonstrated – is a significantly easier task than tool innovation”. While this may be true, it is a bit misleading in the context of the paper as the agent is doing something more like tool selection and modification rather than tool innovation (and actually the in-sample scenes are more like “manufacture”, which the agent doesn’t always even do well on).
>
> A: The point we want to make here is that tool imagination is a step prior to tool manufacture and tool selection. This is because the tool imagination indicates an understanding of the tool affordances and the ability to modify it. Using that knowledge, the agent can then select and manufacture proper tools. Our work demonstrates that this affordance knowledge can be acquired from simple success/failure trials and how it is used for tool imagination. Also, we demonstrated some novel tools in figure 4 and figure 6 in the appendix although they appear less often than the standard sticks and hooks since most of the tasks are designed to be solvable by just hooks and sticks. The agent is trained to only solve the task rather than tries to always choose the more innovative solution.
>
> S: It would be helpful to more clearly explain scene types. Here is some suggested phrasings: in-sample = familiar scenes with familiar tools, interpolation = novel scenes with familiar tools, extrapolation = novel scenes with novel tools. In Table 1 it would be helpful to indicate which scene types are which (in-sample, interpolation, extrapolation).
>
> A: We thank the reviewer for these constructive suggestions, we will clarify these notions in the next iteration.

---

> > ### Comment · AnonReviewer3 · 2019-11-14
> > **Response to authors**
> >
> > Thanks for your detailed responses to my comments. As I mentioned in my review, I do think this is a very interesting and important research direction and I would love to see robust planning for synthesizing tools, so I hope you continue this line of research. However, I still think this particular set of results needs some work.
> >
> > Based on your response (and the paper), it seems to me that you'd like to be able to make two claims:
> >
> > 1. MONet learns about which properties of objects make them useful tools when it is trained to perform a classification task about which one out of multiple tools will solve a task. ("task relevant object affordances are implicitly encoded as directions/trajectories in a structured latent space shaped by experience").
> >
> > 2. Not only does the latent representation encode information about affordances, this information can be effectively used for planning ("the synthesis process modifies emergent, task-relevant object affordances in a targeted and deliberate way").
> >
> > My concerns with these claims are:
> >
> > 1. This is a relatively weak hypothesis. If an agent is trained to perform the classification task, and it can do this task well, what would it look like to *not* have information about object affordances encoded in its latent space? It is interesting that it seems to ignore information such as color, but that does not seem not be the main focus of the paper (and is only shown qualitatively, not quantitatively). If you would like to make this claim more strongly, I think more detailed analysis of the latent space itself is warranted, as would be comparisons to other models which represent the latent space differently.
> >
> > 2. In my opinion, this is the more interesting hypothesis (and the one that the paper seems to be most concerned with). However, it does not seem to me that the experimental results support it. Specifically, the fact that the experimental results are quite weak---for example, that the model performs worse in the case when a feasible tool is already given, and that it does not generalize well even to cases where the tools are familiar---suggests to me in fact that the learned representations are not actually that useful for planning (or at least, not for planning via activation maximization).

---

### Official Review · AnonReviewer1 · 2019-10-22
**Official Blind Review #1**

**Rating:** 3

**Review:**

This paper proposes an algorithm that learns to synthesize tools for the task of reaching. The main idea is to first use unsupervised segmentation algorithm to decompose an image of a set of tools into individual objects; then, with a trained feasibility model, search through the latent space of encoded tools to 'imagine' new ones for reaching.

This paper has clear strengths and weaknesses. It's studying an important problem from a cognitive perspective; it also proposes a novel model for the task, building upon SOTA models. However, the current problem formulation and experiment setup are not well justified, and the experiments are quite limited. I lean toward rejection.

Most importantly, while this paper argues for the importance of an object-centric representation, it conducts most of its search in the pixel space (both as input to the model, and as the output of the imagination). This leads to some unnatural and unphysical results: in the teaser figure, it's true that the final, imagined tool reaches the target; however, the tool itself shouldn't be able to pass the gap/hole on the wall, due to its angular shape. Objects, in essence, have shapes and physical occupancy. Without modeling physics, it's unclear how useful the object-centric representation is.

Imagination is done by searching over the latent space, which limits the model's generalization power to novel tools or new configurations. This is revealed in the results on case H, where the model doesn't work at all.

The experimental results, as just mentioned, are not very impressive, especially given the simplified setup. There are no results on other tasks except reaching. In addition, comparisons with published methods are missing. For example, what if I just train a Pix2Pix from the inputs to successful reaches? That sounds like a reasonable baseline and should be compared with.

Due to all these limitations, I lean toward rejection.

**Experience Assessment:**

I have published in this field for several years.

**Review Assessment: Checking Correctness Of Derivations And Theory:**

I assessed the sensibility of the derivations and theory.

**Review Assessment: Checking Correctness Of Experiments:**

I assessed the sensibility of the experiments.

**Review Assessment: Thoroughness In Paper Reading:**

I read the paper thoroughly.

---

> ### Author Response · Authors · 2019-11-12
> **Response to Reviewer 1**
>
> We would like to thank Reviewer 1 for their review and constructive suggestions. Our responses:
>
> Q: Most importantly, while this paper argues for the importance of an object-centric representation, it conducts most of its search in the pixel space (both as input to the model, and as the output of the imagination). This leads to some unnatural and unphysical results: in the teaser figure, it's true that the final, imagined tool reaches the target; however, the tool itself shouldn't be able to pass the gap/hole on the wall, due to its angular shape. Objects, in essence, have shapes and physical occupancy. Without modeling physics, it's unclear how useful the object-centric representation is.
>
> A: Please see our general comment 1), above. In this work, we demonstrate that the appearance of an object can be planned for specific tasks by exploiting a high-level performance predictor. We agree that a simulated example with physics or even a real-world application of this model would be exciting; but we would also like to believe there is still some merit to the framework presented here.
>
> Q: Imagination is done by searching over the latent space, which limits the model's generalization power to novel tools or new configurations. This is revealed in the results on case H, where the model doesn't work at all.
>
> A: Our work shows that the latent space captures three different kinds of affordances that inherently exist in the dataset: length, width and shape. These task-relevant factors can be modified in a task-driven way. The failure case H shows that a new affordance, i.e. the ‘claw-shape’ required for H, cannot be understood by the agent if it never experienced it before. Inclusion of case H was thus a considered choice to showcase this.
>
> Q: The experimental results, as just mentioned, are not very impressive, especially given the simplified setup. There are no results on other tasks except reaching. In addition, comparisons with published methods are missing. For example, what if I just train a Pix2Pix from the inputs to successful reaches? That sounds like a reasonable baseline and should be compared with.
>
> A: We investigate reaching tasks because recent work [1] in cognitive science shows that crows do not only use tools but also improvise better ones to reach the food in a puzzle box. Our reaching-task design tries to model affordance-learning with a similar experiment, inspired by but not exactly the same as that confronted by crows. To the best of our knowledge, ours is the first work to perform tool-imagination utilising only learning signals from reconstruction and weak supervision from true/false-feasibility of the toolkit given a specific task, and thus were unable to compare with other published baselines. Specifically, in order to compare with a conditional generative model like Pix2Pix we would need paired data of infeasible tools and corresponding feasible ones. Our approach, in contrast, explicitly does not require such alignment.
>
> We could manually add feasible tools to the dataset but that would change the task. Also, human-defined pairs of feasible/infeasible tools would introduce additional (and, we posit, unnecessary) inductive-biases, as compared to the weakly supervising true/false-task feasibility signal that we used. There are many ways in which an infeasible tool might be made feasible. So hand-designing feasible correspondents to infeasible tools would restrict the problem space. In contrast, our approach does not rely on hand-defined correspondence - i.e. it creates novel tools, based on the affordance knowledge gleaned via only unsupervised learning and success/failure signals.
>
> The suggested comparison with conditional VAE models and conditioning activation maximization naturally leads to the question: Why can't we use conditional VAE/GAN models on this problem?
>
> We think the answer is that conditional VAE/GAN models only learn a single generative function which maps condition $x$ to the target $y$, i.e. $p(y \mid z , x)$. Conditioning activation maximization, in contrast, learns a mapping from the condition $x$ to a function $f$ that takes any tool as input and outputs a feasible tool for that conditioned task $x$. This latter case is reminiscent of meta-learning modulo the fact that it is not few-shot. Therefore, conditional VAE/GAN must be trained with stronger supervision.
>
> [1] Bayern, A.M.P.v., Danel, S., Auersperg, A.M.I. et al. Compound tool construction by New Caledonian crows. Sci Rep 8, 15676 (2018) doi:10.1038/s41598-018-33458-z

---

### Author Response · Authors · 2019-11-12
**General comments to the reviewer's questions**

Thanks to the reviewers for their detailed comments.

First, we would like to address a number of misunderstandings and general concerns. Upon reflection, it is clear we could have done better at communicating some of the key ideas in our paper, and we apologise for any confusion this may have caused; thank you for helping us improve on future iterations. With regard to each of the following points, we shall endeavour to articulate our positions more clearly.

1) What is our reaching task?
The task is not about moving an object in a plane, nor is it exactly a navigation task aimed at finding a path from one point to another (although it can certainly be framed this way). Our inspiration was as a ‘puzzle-fitting’ task, involving a robot, standing behind a red line, placing a tool onto the floor from above in such a way that it avoids any blue obstacles while touching the green dot. We note that this top-down view is distinct from threading the tool around or through any obstacles in 2D. Additionally, the task was to find the object that could reach the goal in this way, not to actually find the optimal pose that would result in a successful reach.

2) What do we mean by ‘structured latent space’?
We mean that the distribution of embeddings in the latent space is influenced by architectural design choices and constraints imposed by the network’s loss functions. To be more precise, in latent spaces that have a bottleneck and use Gaussian prior, the semantics represented in the embeddings tend to vary smoothly, as the embeddings themselves change.

3) What do we mean by ‘shaped by experience’?
By ‘experience’ we mean not just the success or failure feasibility targets, but also the toolkit and task inputs. So the generator psi’ learns from the experience of seeing and reconstructing tools, while the performance predictor learns from the ‘experience’ of succeeding or failing at tasks.

4) What do we mean by ‘directions in latent space’?
This term in particular seems to have caused confusion. We do not mean ‘directions’ to denote straight-line vectors in latent space from infeasible tools to feasible tools. Instead our use of this term was as a ‘local direction’. That is, in the region around an embeded point, there appears to be vectors that modify some semantic properties of the embedded tool (again “local directions”). Globally, a better term might be ‘trajectory’ or even ‘path’. More abstractly, we demonstrate that by using a task-based performance predictor to optimise this embedding we can implicitly discover and modify a subset of semantic properties that are useful for task performance while ignoring the others.

In hindsight, some of our writing in the paper can be misunderstood as stating that we see straight-line paths in latent space from infeasible to feasible tools. In situations where we mean or imply the complete path, we will change the word to ‘trajectory’.

5) What is our main contribution?
We demonstrate that (i) task-relevant object affordances are implicitly encoded as trajectories in the latent space and (ii) that these can be leveraged by traversing along a trajectory driven by a task-classifier performing conditioning activation maximization.

Importantly, our work shows that the appearance of an object can be smoothly modulated according to discrete, high-level task descriptions (e.g. a classifier representing task success).

---

### Decision · Program_Chairs · 2019-12-19

**Decision:**

Reject

**Comment:**

This paper investigates the task of learning to synthesize tools for specific tasks (in this case, a simulated reaching task). The paper was reviewed by 3 experts and received Reject, Weak Reject, and Weak Reject opinions. The reviews are very encouraging of the topic and general approach taken by the paper -- e.g. R3 commenting on the "coolness" of the problem and R1 calling it an "important problem from a cognitive perspective" -- but also identify a number of concerns about baselines, novelty of proposed techniques, underwhelming performance on the task, whether experiments support the conclusions, and some missing or unclear technical details. Overall, the feeling of the reviewers is that they're "not sure what I am supposed to get out of the paper" (R3). The authors posted responses that addressed some of these issues, in particular clarifying their terminology and contribution, and clearing up some of the technical details. However, in post-rebuttal discussions, the reviewers still have concerns with the claims of the papers. In light of these reviews, we are not able to recommend acceptance at this time, but I agree with reviewers that this is a "cool" task and that authors should revise and submit to another venue.